# OpenReview forum: "Automatically Eliciting Toxic Outputs from Pre-trained Language Models"
_ICLR.cc/2024/Conference — ICLR 2024 Conference Withdrawn Submission_

### Official Review · Reviewer_osGE · 2023-10-27

**Soundness:** 2 fair
**Presentation:** 2 fair
**Contribution:** 3 good
**Rating:** 5
**Confidence:** 3

**Summary:**

This paper proposes a new optimization algorithm for eliciting toxic outputs from pre-trained language models. The authors demonstrate that their algorithm outperforms other adversarial attack baselines in its efficacy for eliciting toxic content. They also show that their approach can be used to identify and mitigate potential risks associated with the deployment of language models in real-world settings. Overall, the paper's contributions include a new optimization algorithm for eliciting toxic content, a demonstration of its efficacy, and insights into the potential risks and benefits of using such an approach in practice.

**Strengths:**

+ The paper is well-written and clear to read.

+ The idea is novel and the performance improvement is obvious across different settings.

+ Sufficient ablation studies are conducted to support the claim.

**Weaknesses:**

- Limited evaluation: The authors only evaluate their approach on a single dataset. This could limit the generalizability of their results and make it difficult to draw broader conclusions about the effectiveness of their approach.

- Limited scope: The paper only focuses on the generation of toxic content from language models and does not address other potential risks associated with their deployment, such as bias or misinformation. This could limit the paper's relevance and impact in the broader context of language model research.

- Additionally, the paper only tests on 1/2/3-word output, which is not practical. It is unknown whether the proposed method would work for longer toxic outputs.

**Questions:**

NA

**Details Of Ethics Concerns:**

I personally do not like the motivation of this paper, i.e., designing methods to elicit toxic content from LLMs. This could lead to bad results if no proper defense methods are developed. Given the fast spread of LLMs today, such algorithms could involve potential ethical concerns.

---

> ### Author Response · Authors · 2023-11-22
> **Official Response to Reviewer osGE by Authors**
>
> Thanks for your constructive comments and suggestions! We hope our response resolves your concerns.
>
> ## Weakness1
>
> As pre-trained language models advance in many tasks, addressing safety concerns becomes increasingly necessary and imperative. Our research explores the potential risk of publicly available language models and critically assesses their vulnerability. We believe that AI research always comes with social and ethical issues, and our work might contribute to the enhancement of language model safety in the future. We have supplemented an ethnical statement section in our paper. We really appreciate the suggestion!
>
> We have carried out experiments on another sentence dataset and the results are reported in the response to all reviewers.
>
> ## Weakness2
>
> As pre-trained language models are seeing increasing use across a wide variety of applications, a lot of problems in language models have been discovered and studied, e.g. toxicity, hallucination, bias, etc. Due to the different sources of these problems in language models, it is very difficult to study them under the same framework. Most existing researches also view these problems as different vulnerabilities in language models and study them separately. In this paper, we mainly focus on the toxicity and security risks of language models.  Bias like gender and racial discrimination is also included in some toxic expression.
>
> We have tried to automatically construct prompts to generate statements that violate common sense such as examples in [1] without changing parameters. However, all existing algorithms do not work well in this task. The possible reason is that toxic outputs of language models output are due to the toxic examples in the training data [2], while misinformation and hallucinations can be caused by architecture flaw and suboptimal training objective [3]. Therefore, most current researches divide these potential risks into different types for study.
>
> As discussed in Section 5.2 in our paper, the success rate of the attack is strongly correlated with the perplexity of the target output. In other words, our method can achieve a good performance if the target text has high fluency on the language model.
>
> We think it is meaningful to propose a framework that can elicit different kinds of potential risks at the same time. We will further study this problem in future work.
>
>
>
> ## Weakness3
>
> Following established experimental settings of adversarial prompts [4], we construct test dataset containing 1/2/3 tokens from Civil Comments Dataset. The adaptation of the tokenization method for the test data is performed to align it with the requirements of various language models. As longer output text requires a longer prompt to elicit, relatively short test data help reduce the computational cost of the optimization. To better alleviate your concerns, we have carried out experiments on a new sentence dataset to alleviate your concerns. The experimental results are reported in the response to all reviewers.
>
>
>
> [1]：Meng, Kevin, et al. "Locating and editing factual associations in GPT." *Advances in Neural Information Processing Systems* 35 (2022): 17359-17372.
>
> [2]：Nicholas Carlini, Florian Tramer, Eric Wallace, Matthew Jagielski, Ariel Herbert-Voss, Katherine Lee, Adam Roberts, Tom Brown, Dawn Song, Ulfar Erlingsson, et al. Extracting training data from large language models. In 30th USENIX Security Symposium (USENIX Security 21), pp. 2633–2650, 2021.
>
> [3]：Huang, Lei, et al. "A survey on hallucination in large language models: Principles, taxonomy, challenges, and open questions." *arXiv preprint arXiv:2311.05232* (2023).
>
> [4]：Jones, Erik, et al. "Automatically Auditing Large Language Models via Discrete Optimization." *arXiv preprint arXiv:2303.04381* (2023).

---

> > ### Author Response · Authors · 2023-11-23
> > **Official Comment by Authors**
> >
> > Dear Reviewer osGE：
> >
> > We wonder if our response answers your questions and addresses your concerns.
> >
> > Thanks again! Sincerely looking forward to your reply!
> >
> > the authors

---

### Official Review · Reviewer_QEDL · 2023-10-30

**Soundness:** 3 good
**Presentation:** 3 good
**Contribution:** 3 good
**Rating:** 6
**Confidence:** 3

**Summary:**

With the emergence of LLMs, it is crucial to discover and modify potential toxic outputs before deployment.
In this work, authors propose ASRA, a new optimization algorithm that concurrently updates multiple prompts and selects
prompts based on determinantal point process. Experimental results on six
different pre-trained language models demonstrate that ASRA outperforms other
adversarial attack baselines in its efficacy for eliciting toxic content.

**Strengths:**

* The experiments show significant perfanmance improvement over previous works.
* The authors provide detailed and insightful analysis about hyperparameter tuning and future application.

**Weaknesses:**

* The proposed method seems a little complicated, especially the DPP procedure.  I'd like to know it's speed (e.g. throughput ) compared with
previous works.
* Experiments limits on eliciting toxic text of up to 3 words. In practice, however, what we really care about is not generating
toxic text at any length. Therefore I have a concern about this work's transferability to a more practical scenario.

**Questions:**

* I want to know that could DPP be possibly replaced by other simpler method, e.g. methods often used in document summarization task?

---

> ### Author Response · Authors · 2023-11-22
> **Official Response to Reviewer QEDL by Authors**
>
> Thanks for your constructive comments and suggestions! We hope our response resolves your concerns.
>
> ## Weakness1
>
> We adopt DPP to model the negative correlations between quality and diversity in beam selection. Calculating the determinant of the kernel matrix precisely requires a high computational cost, previous algorithm that provide exact implementation has *O($M^4$)* complexity, where $M$ denotes the total number of candidate items [1]. However, the approximate greedy algorithm used in our implementation has *O($N^2M$)* complexity when selecting $N$ items out of $M$ [2]. Therefore, actually in ASRA, the running time of DPP selection procedure is extremely short and almost negligible. We have calculated the average running time in our datasets. One iteration takes around  0.732s on GPT-J and 0.834s on LLaMA in total, while the DPP algorithm only takes 0.002s on these language models. We provide the average running time of one iteration in the algorithm and the corresponding throughput of different methods on LLaMA  in the following table:
>
> |                         | GBDA  | AutoPrompt | ARCA  | ASRA (-) DPP | ASRA (+) DPP |
> | :---------------------: | :---: | :--------: | :---: | :----------: | :----------: |
> | Running Time per iter/s | 0.185 |   0.203    | 0.283 |    0.832     |    0.834     |
> |  Throughput (token/s)   | 48.65 |   44.33    | 31.80 |    10.82     |    10.79     |
>
>
>
> ## Weakness2
>
> Given any length of toxic text, our algorithm will search for prompt to elicit it from the language model. Obviously, longer toxic text is harder to elicit from language models, thus requires a sequence with more tokens to be optimized as the prompt. As prompt length increases, all adversarial algorithms face a higher computational cost.
>
> Following established experimental settings of adversarial prompts [3], we construct test dataset containing 1/2/3 tokens from Civil Comments Dataset. The adaptation of the tokenization method for the test data is performed to align it with the requirements of various language models. As longer output text requires a longer prompt to elicit, relatively short test data help reduce the computational cost of the optimization.
>
> To better alleviate your concerns, we have carried out experiments on a new sentence dataset to alleviate your concerns. The experimental results are reported in the response to all reviewers.
>
> ## Question
>
> DPP model is used to model the negative correlation between quality and diversity to avoid redundant search in the algorithm. This property of algorithm is very important in solving the task. We make attempts on a famous algorithm for unsupervised summarization, Textrank [4]. The success rate of the algorithm on LLaMA is provided as follows.
>
> |                | (+)Textrank | (+)DPP |
> | -------------- | ----------- | ------ |
> | 1-word dataset | 6.85%       | 91.78% |
> | 2-word dataset | 1.27%       | 36.02% |
> | 3-word dataset | 0%          | 10.71% |
>
> The algorithm which mainly models similarity between different candidates to select items does not work well in this task.
>
>
>
> [1]：J. Gillenwater, A. Kulesza, and B. Taskar. Near-optimal MAP inference for determinantal point
> processes. In Proceedings of NIPS 2012, pages 2735–2743, 2012.
>
> [2]：Laming Chen, Guoxin Zhang, and Eric Zhou. Fast greedy map inference for determinantal point process to improve recommendation diversity. Advances in Neural Information Processing Systems, 31, 2018
>
> [3]：Jones, Erik, et al. "Automatically Auditing Large Language Models via Discrete Optimization." *arXiv preprint arXiv:2303.04381* (2023).
>
> [4]：Mihalcea, Rada, and Paul Tarau. "Textrank: Bringing order into text." *Proceedings of the 2004 conference on empirical methods in natural language processing*. 2004.

---

> > ### Comment · Reviewer_QEDL · 2023-11-22
> > **Reply to Rebuttal**
> >
> > Thanks for your clarification and additional experiments in your rebuttal, which has solved most of my concerns. I'll increase my score to 6.

---

### Official Review · Reviewer_xui8 · 2023-10-30

**Soundness:** 3 good
**Presentation:** 3 good
**Contribution:** 3 good
**Rating:** 6
**Confidence:** 3

**Summary:**

The paper presents a reversing PLM approach to generate prompts that lead to toxic outputs. The proposed method consists of an approximation step, a refinement step, and a selection step implemented via a DPP model. Empirical studies with 6 PLMs indicate the effectiveness of the proposed approach.

Overall, this is an interesting paper with a decent algorithm and compelling results. Minor concerns:

(1) The evaluation sets seem small (a few hundreds of examples), is it possible to report some evaluation results on large datasets?

(2) Normally, it is more important to detect prompts without any malevolent words but leading to toxic responses for a PLM. However, from the cases shown in the Appendix, the prompts found by the proposed method usually contain malevolent words. This may limit the application of the proposed method in practice. Then, is it possible to further improve the method by avoiding malevolent words when reversing PLMs?

(3) Since the algorithm needs to search every word in the vocabulary, when the vocabulary size is big, will the complexity becomes an issue in application of the method?

**Strengths:**

decent algorithm
compelling results

**Weaknesses:**

small evaluation sets
complexity could be an issue

**Questions:**

see the summary

---

> ### Author Response · Authors · 2023-11-22
> **Official Response to Reviewer xui8 by Authors**
>
> Thanks for your constructive comments and suggestions! We hope our response resolves your concerns.
>
> ## Weakness1
>
> Following established experimental settings of adversarial prompts [1], we construct test dataset containing 1/2/3 tokens from Civil Comments Dataset. The adaptation of the tokenization method for the test data is performed to align it with the requirements of various language models. As longer output text requires a longer prompt to elicit, relatively short test data help reduce the computational cost of the optimization.
>
> We have carried out experiments on a new sentence dataset to alleviate your concerns. The experimental results are reported in the response to all reviewers.
>
> ## Weakness2
>
> By introducing a BERT-based toxicity classifier, we compute the unigram toxicity score of each token in the vocabulary. We force the model not to generate tokens with relatively high toxicity scores, containing around 3, 000 tokens. We call this the toxicity constraint. The performance of different algorithms with and without the toxicity constraint is illustrated in the table below. We perform experiments on GPT-2 and GPT-J.
>
> | Dataset     | Constraint         | GPT-2  | GPT-J  |
> | ----------- | ------------------ | ------ | ------ |
> | 1-word data | (-) tox_constraint | 97.26% | 97.26% |
> | 1-word data | (+) tox_constraint | 98.63% | 94.52% |
> | 2-word data | (-) tox_constraint | 69.49% | 63.14% |
> | 2-word data | (+) tox_constraint | 63.98% | 61.02% |
> | 3-word data | (-) tox_constraint | 23.36% | 27.49% |
> | 3-word data | (+) tox_constraint | 20.92% | 21.41% |
>
> As illustrated in the table, the toxicity constraint doesn't have a large impact on the performance of our algorithm.
>
>
>
> ## Weakness3
>
> ASRA calculates an approximate probability for each token in the vocabulary in step 1. The approximate value is computed with the average first-order approximation value. As we discuss in Section 3 in our paper, the approximation depends on $t$ random tokens and can be computed efficiently with one gradient back propagation and matrix multiplication between the gradient matrix and the embedding table matrix. The matrix multiplication has a complexity $O(ld_{e}|\mathcal{V}|)$, where $l$ denotes the sequence length, $d_e$ denotes the dimension of word embedding and $|\mathcal{V}|$ denotes the size of the vocabulary. We have recorded the computational time of this process in each iteration. The approximation step occupies 0.62s of in the experiments on GPT-2 with a vocabulary size of 50,257. Therefore, although the time consumption increases as the vocabulary size becomes larger, the algorithm can still be applied in practical scenario.
>
> [1]：Jones, Erik, et al. "Automatically Auditing Large Language Models via Discrete Optimization." *arXiv preprint arXiv:2303.04381* (2023).

---

> > ### Author Response · Authors · 2023-11-23
> > **Official Comment by Authors**
> >
> > Dear Reviewer xui8：
> >
> > We wonder if our response answers your questions and addresses your concerns.
> >
> > Thanks again! Sincerely looking forward to your reply!
> >
> > the authors

---

### Official Review · Reviewer_crro · 2023-11-06

**Soundness:** 3 good
**Presentation:** 3 good
**Contribution:** 2 fair
**Rating:** 5
**Confidence:** 3

**Summary:**

This paper discusses the challenge of preventing pre-trained language models from generating toxic content. The authors focus on the process of "reversing" language models to intentionally produce such content, which serves as a test to identify and mitigate potential risks before deployment. They present ASRA (Auto-regressive Selective Replacement Ascent), an optimization technique designed to efficiently elicit toxic output from language models. ASRA works by updating multiple prompts simultaneously and using a determinantal point process to select the most effective ones. The algorithm was tested on six different language models, and the results showed that ASRA surpasses other adversarial attack methods in terms of eliciting toxic content. Additionally, the research found a significant link between the success of ASRA and the perplexity of the generated output, with less connection to the language models' size. The findings suggest that reversing language models with a comprehensive dataset of toxic text can be a strategic method for evaluating and improving the safety of language models.

**Strengths:**

- This paper proposes a novel method of textual adversarial attack for generation tasks, which looks generalizable for different text generation tasks.
- The empirical results look convincing, outperforming the baselines by a large margin
- This paper focuses on AI safety, which is an important topic recently regarding safeguarding the output of LLMs. The authors offer a potential tool for developers to identify and fix vulnerabilities before deployment.

**Weaknesses:**

- Lack of baselines: The attack algorithm is based on HotFlip (2018), which is a bit old and less effective than the recently proposed baselines. I am wondering if the authors have compared with adversarial attack baselines proposed more recently such as Seq2sick [1], which shows better optimization effectiveness for text-to-text generation tasks.
- Lack of defense models (detoxified models): While I appreciate the authors’ efforts in comparing different pretrained models, it would also be interesting to evaluate against different defense approaches/detoxification approaches, such as [2,3,4], and confirm whether the attack is still effective.
- Validity of toxicity evaluation: The authors mention that their evaluation setup “can be applied to bridge the evaluation of toxicity in different PLMs”, and “speculate that the success rate of ASRA attack might be positively correlated with the toxicity of language models.” However, I do not see any evidence about whether the ASR here can be a good proxy to reflect model toxicity. Given that the model toxicity evaluation is conducted by evaluating model responses with a lot of different inputs and contexts, the setup of this paper is to evaluate model responses given “unnatural” prompts. I am thus suspicious about whether the test above can give an accurate evaluation of model toxicity.
- There is an important concern regarding the potential misuse of this work by malicious users to bypass safety controls of LLMs and elicit model toxicity. I believe it would be valuable for the paper to include a discussion of this aspect.

[1] Seq2Sick: Evaluating the Robustness of Sequence-to-Sequence Models with Adversarial Examples. (AAAI 2020)
[2] Plug and Play Language Models: A Simple Approach to Controlled Text Generation (ICLR 2020)
[3] DExperts: Decoding-Time Controlled Text Generation with Experts and Anti-Experts (ACL 2021)
[4] Exploring the Limits of Domain-Adaptive Training for Detoxifying Large-Scale Language Models (NeurIPS 2022)

**Questions:**

The proposed optimization seems generalizable for different text generation tasks. Have you ever used the approach to attack other text generation tasks (such as [1])?

**Details Of Ethics Concerns:**

There is an important concern regarding the potential misuse of this work by malicious users to bypass safety controls of LLMs and elicit model toxicity. I believe it would be valuable for the paper to include a discussion of this aspect.

---

> ### Author Response · Authors · 2023-11-22
> **Official Response to Reviewer crro by Authors**
>
> Thanks for your constructive comments and suggestions! We hope our response resolves your concerns.
>
> ## Weakness1
>
> In our paper, we choose GBDA, Autoprompt and ARCA as baselines. Although Hotflip is an old method in adversarial attack, latter algorithms have made optimizations on the basis of Hotflip which achieve a big leap in performance. Notably, methodologies such as Seq2Sick [1] introduce subtle perturbations to input sequences, thereby disrupting the efficacy of language models across diverse tasks. The primary objective of this algorithm is assessing the robustness of language models through the introduction of such perturbations.
>
> However, the motivation of the algorithm is to deviate language models from the correct output to construct adversarial samples in various text generation tasks. In our task,  we want to elicit the language model to produce specific content that is considered as toxic or unsafe. The optimization objectives of these two categories of algorithms are inherently distinct. As a result, direct comparisons within the confines of the same experimental setting is challenging.
>
> Therefore, we add a discussion about this algorithm in the Related Work section instead. We really appreciate the suggestion!
>
>
>
> ## Weakness2
>
> We supplement the performance of our algorithm against two defense approaches, including DAPT [2] and DExperts [3]. We follow the implementation in the original paper and test the performance of our algorithm against the two defense approaches on GPT-2. The results are listed as follows:
>
> | Dataset     | GPT-2  | DAPT   | DExperts |
> | ----------- | ------ | ------ | -------- |
> | 1-word data | 97.26% | 95.89% | 82.19%   |
> | 2-word data | 69.49% | 70.76% | 18.22%   |
> | 3-word data | 23.36% | 28.71% | 8.27%    |
>
> According to the experimental results, ASRA can also elicit some toxic cases even against defense approaches. Generally, DAPT does not improve the robustness of the pre-trained language model against our adversarial attack, while DExperts achieves a relatively good performance. The attack and evaluation of defense methods need further research in the future.
>
>
>
> ## Weakness3
>
> Most existing toxicity evaluation is conducted by evaluating model responses with human-written inputs. The evaluation method of safety of pre-trained language models depends on the practical scenario. Common evaluation assumes that users query language models with some questions or dialogue. However, malicious attackers can attack pre-trained language models by intentionally constructing adversarial prompts. Ensuring the safety of language models requires evaluation at different levels. This paper explores how our algorithm can assess the safety of pre-trained language models when facing intentional adversarial challenges.
>
>
>
> ## Weakness4
>
> We agree with your concern that our approach might be used by malicious attackers to attack public large pre-trained language models, leading to toxic content generation or privacy leakage. We believe that AI research always comes with social and ethical issues, and our work might contribute to the enhancement of language model safety in the future. We have supplemented an ethnical statement section in our paper. We really appreciate the suggestion!
>
>
>
> ## Question
>
> Our algorithm is not specific to text generation tasks like summarization, but to test the security and toxicity of pre-trained language models. The objective of our algorithm is to elicit certain content from the language model. Therefore, it is challenging to conduct experiments on various text generation tasks like [1].  As we discuss in Section 5.2, the success rate of the attack is strongly correlated with the perplexity of the target output. In other words, our method can achieve a good performance if the target text has high fluency on the language model.
>
>
>
> [1]：Seq2Sick: Evaluating the Robustness of Sequence-to-Sequence Models with Adversarial Examples.
>
> [2]：Exploring the Limits of Domain-Adaptive Training for Detoxifying Large-Scale Language Models
>
> [3]：DExperts: Decoding-Time Controlled Text Generation with Experts and Anti-Experts

---

> > ### Author Response · Authors · 2023-11-23
> > **Official Comment by Authors**
> >
> > Dear Reviewer crro：
> >
> > We wonder if our response answers your questions and addresses your concerns.
> >
> > Thanks again! Sincerely looking forward to your reply!
> >
> > the authors

---

### Author Response · Authors · 2023-11-22
**Official Response to All Reviewers by Authors**

Thanks for your constructive comments and suggestions! We hope our response resolves your concerns.

Following established experimental settings of adversarial prompts [1], we construct test dataset containing 1/2/3 tokens from Civil Comments Dataset. The adaptation of the tokenization method for the test data is performed to align it with the requirements of various language models. As longer output text requires a longer prompt to elicit, relatively short test data help reduce the computational cost of the optimization.

To alleviate your concerns that our algorithm cannot work well in a practical scenario, we have extracted a dataset consisting of succinct yet complete toxic sentences. Cases in the dataset exhibit enhanced sentence structures and greater length. To balance the algorithmic performance and computational efficiency, we choose a prompt length = 8 in this experiment on LLaMA. The performance of different algorithms are shown as follows:

|                        | GBDA | AutoPrompt | ARCA  | ASRA (-) DPP | ASRA (+) DPP |
| ---------------------- | ---- | ---------- | ----- | ------------ | ------------ |
| sentence dataset | 0%   | 1.69%      | 3.39% | 11.86%       | 20.34%       |

Our algorithm achieves the best performance in the new dataset as well, and the effect of the DPP module is more obvious.

[1]：Jones, Erik, et al. "Automatically Auditing Large Language Models via Discrete Optimization." *arXiv preprint arXiv:2303.04381* (2023).

---

### Meta-Review · Area_Chair_iNEd · 2023-12-06

**Metareview:**

The authors present ASRA (Auto-regressive Selective Replacement Ascent), an optimization technique that efficiently elicits toxic output from PLMs. ASRA operates by simultaneously updating multiple prompts and employing a determinantal point process for selecting the most effective ones. Experiments conducted on six distinct PLMs demonstrate that ASRA outperforms other adversarial attack methods in terms of eliciting toxic content.

**Pros:**

* Novel and effective method (ASRA) for eliciting toxic content from PLMs.
* Outperforms existing adversarial attack methods.
* Contributes to AI safety by providing a tool for identifying and mitigating potential risks before deployment.
* Comprehensive evaluation across multiple language models.

**Cons:**

* Limited evaluation raising concerns about generalizability, reliability.
* Narrow focus on generating toxic content without addressing other potential risks associated with PLM deployment.
* Lack of comparison with recent adversarial attack baselines and defense models.
* Validity of toxicity evaluation is questionable, as the method relies on unnatural prompts.

**Justification For Why Not Higher Score:**

The paper can be much stronger by addressing few key concerns from the reviewers:
- Compare ASRA with more recent baselines
- Conduct a more comprehensive evaluation of the validity of the toxicity evaluation.
- Discuss the potential for misuse of the research.
- Explore methods for avoiding malevolent words in the prompts.

**Justification For Why Not Lower Score:**

N/A

---

### Decision · Program_Chairs · 2024-01-16

Reject